



# Different sensitivities of litter decomposition and nutrient release to ultraviolet radiation

Weiming Yan[1,2]*, Zhouping Shangguan[1], Yangquanwei Zhong[3]*

[1] State Key Laboratory of Soil Erosion and Dryland Farming on the Loess Plateau, Northwest A&F University, Yangling, Shaanxi 712100, P.R. China

[2] Institute of soil and water conservation, Chinese Academy of Sciences, Yangling, Shaanxi 712100, P.R. China

[3] Center for Ecological and Environmental Sciences, Key Laboratory for Space Bioscience & 10 Biotechnology, Northwestern Polytechnical University, Xi'an, 710072, P.R. China

*Correspondence to*: zhongyqw@foxmail.com

**Abstract.** Ultraviolet (UV) radiation plays an important role in litter decomposition. Despite years of research, it is still not fully understood that the role of UV radiation in litter decomposition and carbon

(C) and nutrient release, as well as the direct (litter decay directly exposed to UV) and indirect (plants received UV during growth) effects. In this study, a meta-analysis that comprised 54 published studies concerning UV radiation experiments, including 598 observations, was performed to quantify the responses of litter decomposition and C and nutrient release to UV radiation. The direct and indirect effects of UV-B enhancement on litter decomposition did not differ, but the direct effects of litter

weight loss were more sensitive than the indirect effects under UV attenuation. Changes in UV radiation did not affect litter decomposition under laboratory conditions, but litter type only affected the rate of decomposition. In addition, the weight loss under UV radiation exhibited three-stage temporal dynamics: UV enhancement accelerated litter decomposition but required UV accumulation, whereas the UV attenuation effect decreased with time. The litter decomposition rates increased with

decreasing precipitation (<450 mm), and the sensitivity increased as the precipitation decreased (<700 mm). Overall, UV attenuation showed considerable effects on C and nitrogen (N) release. The findings of this meta-analysis suggested that photodecomposition effects on litter decomposition depend on many factors and can alter C and N cycling during decomposition.

**Keywords:** Ultraviolet radiation, direct/indirect effects, litter types, weight loss, nutrient release



### Introduction

Plant litter decomposition plays a key role in carbon (C) and nutrient cycling in terrestrial ecosystems (García‑Palacios et al., 2016;Prescott, 2005;Xu et al., 2013;Zhang et al., 2013). Increasing numbers of studies have proven the importance of abiotic (e.g., warming and drought, solar ultraviolet (UV) radiation) and biotic (e.g., plant species and diversity, soil decomposer communities) drivers of litter decomposition (Almagro et al., 2017;King et al., 2012;Brandt et al., 2010). As the key driver of C fixation and energy flow in global ecosystems, UV radiation plays an important role in plant growth and C cycling (Chapin III et al., 2011), and has attracted the attention of many researchers. In addition, the results of extensive research have shown that UV radiation is also an important driver of litter decomposition in ecosystems, although the magnitude and direction of decomposition differ among studies (Day et al., 2015;Brandt et al., 2009;Brandt et al., 2010). Due to the increase in human activities, the amount of UV radiation reaching the earth's surface has changed (Williamson et al., 2014), including increased UV radiation in the Southern Hemisphere (Herman, 2010) and reduced UV radiation in the Northern Hemisphere (Calbo and González, 2005). Consequently, research on the effects of UV radiation on litter decomposition and C and nutrient release could provide important crucial information for the prediction of C and nutrient cycling in terrestrial ecosystems in the context of changes in UV radiation.

In general, soil microbe plays key role in litter decomposition (Moorhead and Sinsabaugh, 2006). However, in some regions, litter decomposition is proportional to time and cannot be explained by exponential decomposition models (Parton et al., 2007). Under the circumstances, UV radiation is recognized as one of the most important drivers of litter decomposition responsible for the unexpected rapid decomposition, which has been proven in arid ecosystems (Austin and Ballare, 2010;Brandt et al., 2009;Day et al., 2015). The effect of UV radiation on litter decomposition can be divided into two main processes: first, UV radiation can break down organic matter directly into C-based gases (Brandt et al., 2009;Lee et al., 2012;Rutledge et al., 2010); second, UV radiation can facilitate the conversion of large resistant compounds to smaller compounds that are more readily degradable by soil microbes (Lambie et al., 2014;Gallo et al., 2009).

The effect of UV radiation on litter decomposition also varies under different precipitation regimes,





and previous studies have proven the positive effects of UV radiation on litter decomposition in areas with an annual precipitation ranging from 152–726 mm (Pancotto et al., 2003;Day et al., 2015;Huang and Li, 2017;Huang et al., 2017). In areas of lower precipitation, although UV radiation inhibits microbial decomposition, photodecomposition is still a dominant factor in litter decomposition because the little

amount of vegetation causes the litter to receive more solar radiation (Austin and Vivanco, 2006), which directly increases the rate of organic matter breakdown and could supply easily decomposable substrates to soil decomposers (Austin and Vivanco, 2006;Pancotto et al., 2003;Brandt et al., 2010;Huang et al., 2017). However, the effects of UV radiation on litter decomposition have often shown inconsistent results under dry and wet conditions. For example, greater effects on litter decomposition have been reported

under dry conditions than under wet conditions (Brandt et al., 2007); in contrast, other studies have reported that the effects of UV radiation on litter decomposition under dry conditions are negligible (Uselman et al., 2011). Litter decomposition under conditions of low precipitation is the result of the balance between positive photodecomposition and negative biodecomposition due to low amounts of available water. However, the size effect of changes in UV radiation on litter decomposition under

different precipitation regimens remains unclear, which is important for understanding C and nutrient cycling.

Many studies have demonstrated that UV radiation can affect litter decomposition and can alter C and nutrient cycling among plants, the soil and the atmosphere, although the mechanisms, direction and magnitude differ among these studies (Brandt et al., 2010;King et al., 2012;Gallo et al., 2009). Litter

decomposition may be affected directly or indirectly by UV radiation and involves both biotic and abiotic processes. The direct effects of UV radiation include altering the decomposition rate directly via the photochemical breakdown of the litter or altering the abundance and community composition of decomposers. The indirect effects of UV radiation are manifested as changes in the chemical composition and physical properties of the litter during the growth and senescence of plants (Wang et al., 2015).

Therefore, studies on the effects of UV radiation on litter decomposition have shown inconsistent responses due to different response patterns (Pancotto et al., 2003). Experimental results of the effects of UV enhancement on plants have shown increases, no effect or even decreases in litter decomposition (Newsham et al., 2001;Hoorens et al., 2004;Song et al., 2013b), as well as the experimental results of the effects of UV enhancement on the soil (Moody et al., 2001;Gehrke et al., 1995). In addition, the direction

and magnitude of litter decomposition under changing UV conditions also depend on plant species,



vegetation type, decay period length and experimental conditions (Song et al., 2013a;Kirschbaum et al., 2011). For example, Pancotto et al. (2005)) reported that UV attenuation on the soil reduced litter decomposition but increased it when plants were grown under conditions of UV attenuation. However, the effects of UV radiation on litter decomposition showed the opposite results after different durations

(Song et al., 2013b), and research under controlled laboratory conditions has also demonstrated no effect of   UV enhancement on litter decomposition (Kirschbaum et al., 2011). Together, all these factors limit our generalization of the mechanism of the effects of UV radiation on litter decomposition. Thus, to better understand the role of UV radiation on litter decomposition, different factors such as vegetation type and experimental duration and incubation conditions should be considered.

Previous studies on the effects of UV radiation on litter decomposition have concentrated mainly on the loss of litter weight. Less attention has been paid to the release of C and nutrients from litter or on the correlation between litter weight loss and nutrient release in response to changes in UV radiation (Wang et al., 2015). Two meta-analyses related to the effect of UV radiation on litter decomposition have been conducted (Wang et al. 2015; Song et al. 2013a). However, the results of these meta-analyses varied

widely because of differences in the compiled data sets. One of these studies mainly emphasized the litter weight remaining and its chemistry under elevated UV radiation (Wang et al. 2015), while the other only examined the litter weight remaining under changes in UV-B radiation (Song et al. 2013a), and the datasets used in both studies were small (81 and 93 UV radiation experiments). In general, the loss of litter weight increases as decomposition time increases, but nutrient release exhibits different results. For

example, the nitrogen (N) remaining in litter was shown to increase after fifteen months photodegradation of litter decomposition in semiarid Mediterranean grasslands (Almagro et al., 2017). Thus, to better understand the C and nutrient release from litter, clarifying the correlation between weight loss and nutrient release during litter decomposition under changes in UV radiation is urgently needed.

To clarify the uncertainty on the effects of UV radiation on litter decomposition, and specifically

the role of the effects of UV radiation on C and nutrient release in the litter decomposition process, we conducted a meta-analysis of studies based on litter decomposition worldwide using UV radiation. Our main goal was to resolve the conflicting results presented to date and to clarify the response of nutrient release to UV radiation, which may be different from the rate of litter weight loss. A total of 598 datasets were collected, and we addressed the following: (1) how litter decomposition and C and nutrient release

respond to UV radiation under different experiment conditions (UV radiation types, direct or indirect

effects of UV radiation, experimental conditions, litter type and experiment duration); (2) how the sensitivity of litter decomposition rates changes in response to UV radiation in different climate areas; and (3) whether the relationship between C and nutrient release and litter weight loss changes under changes in UV.

**1    Materials and Methods**

**2.1 Data preparation**

Published articles were identified using the Web of Science and online databases of the Chinese Academy of Sciences (prior to December 2017) by querying the following combinations of terms: (ultraviolet/UV/photodecomposition/UV-B) and (litter decomposition/litter quality/litter nutrients). To

avoid bias in the selection of publications, articles were selected based on the following criteria: (1) the study included at least one paired data set (control and treatment) from experiments involving UV enhancement and attenuation as well as and photodecomposition; (2) weight loss and remaining nutrients in litter measured after different durations were denoted separately; and (3) the mean, standard deviation/error and number of replicates in the control and treatment groups could be calculated or

directly extracted from the text, tables or digitized graphs.

For each selected study, the experimental location and environmental variables, such as the mean annual temperature and mean annual precipitation, were obtained directly from published papers. In addition, UV radiation types, the indirect or direct effects of UV radiation, plant litter species, litter type (herb or wood), initial litter weight, and litter chemical properties were also recorded. In total, 54

published papers covering global sites (Fig. S1) that satisfied our selection criteria for this study were selected from more than 1000 published papers. The list of the literature sources and data are shown in the Supporting Information. All original data were extracted from the text, tables, figures and appendices of those publications. For the studies whose data were presented graphically, Get-Data Graph Digitizer (ver. 2.20, Russian Federation) was used to digitize and extract the numerical data.

**2.2 Data analysis**

The response ratios (RRs, the natural logs of the ratios of the mean values of the parameters in the treatment group to those in the control group) for the biomass loss and nutrient changes were evaluated





using the following equation (Hedges et al., 1999):

$$RR = \ln(Xe / Xc) = \ln Xe - \ln Xc \qquad \text{Eq. (1)},$$

where Xe and Xc are the response values of each individual observation in the experimental and control

treatments, respectively. The corresponding sample variance for each RR was calculated as follows:

$$vi = (Se / Xe)^2 / ne + (Sc / Xc)^2 / nc \qquad \text{Eq. (2)},$$

where ne, Se, and Xe represent the sample size, standard deviation and mean response values in the

experimental group, respectively, and nc, Sc, and Xc represent the sample size, standard deviation and

mean response values in the control group, respectively. The reciprocal of the variance (w = 1 / vi) was

considered the weight of each RR. The mean weighted response ratio (RR$_{++}$) was calculated from the RR

for individual pairwise comparisons between the treatment and control groups as follows:

$$RR_{++} = \sum_{i=1}^{m} \sum_{j=1}^{k} w_{ij} \, RR_{ij} / \sum_{i=1}^{m} \sum_{j=1}^{k} w_{ij} \qquad \text{Eq. (3)},$$

where m is the number of groups and k is the number of comparisons in the corresponding group. In

addition, the standard error of RR$_{++}$ was estimated as follows:

$$(RR_{++}) = \sqrt{\frac{1}{\sum_{i=1}^{m} \sum_{j=1}^{k} w_{ij}}} \qquad \text{Eq. (4)}.$$

In the UV enhancement dataset, only UV-B enhancement was included, whereas the UV attenuation

dataset included both UV-B and UV-(A+B) attenuation. A meta-analysis was performed using the R

software package (version 3.1.1) (R Core Team, 2014). The natural logs of the RRs for the individual

and combined treatments were determined by specifying the studies as random factors in the model with

the "metafor" package. The effects of changes in UV radiation on the loss of biomass and nutrient

changes were considered significant if the 95% confidence interval (CI) of the RR did not overlap with

zero. The "maps" package was used to generate a map of the global site distribution (Fig. S1) (Becker

and Wilks, 2005). The meta-analytic models were selected using the same approach as that used by Terrer

et al. (2016) and van Groenigen et al. (2017)), in which all possible models that could be constructed

using combinations of the experimental factors (changes in UV radiation, litter type, experiment type,

direct/indirect effects, incubation time) described above were considered main effects, using the "glmulti"

package in R. The relative importance of the factors was then calculated as the sum of the Akaike weights

derived for all the models in which the factor occurred. A regression analysis was conducted to evaluate

the relationship between the nutrient release and loss of litter weight, and a general linear model was

used to compare the slopes of the loss of litter weight and nutrient release between the UV treatment and

control groups.

## 2   Results

### 3.1 UV radiation types

As expected, the UV enhancement and attenuation showed opposite effects on the weight loss and
nutrient changes. The UV-B enhancement and attenuation showed significant effects on weight loss, with
RRs of 0.04 and -0.43 (Fig. 1), respectively; the UV-(A+B) attenuation also significantly decreased the
weight loss but showed little effect compared with UV-B attenuation. In addition, UV-B enhancement
and attenuation showed significant effects on k decay, with greater RRs than that for the weight loss.
Both UV-B enhancement and attenuation showed no effect on C and lignin release, which showed
opposite effects on P release.

### 3.2 Direct and indirect effects

The sum of Akaike weights indicates that the litter decomposition was affected by the UV
enhancement or attenuation, litter types (herb or wood), experimental conditions (field or laboratory),
direct (litter decay directly exposed to UV) or indirect (plants received UV during growth) effects, and
duration (Fig. S2). These results were then used to calculate the treatment effects for the changes in each
experimental condition. Due to the few differences between UV-B and UV-(A+B) attenuation on litter
decomposition, we combined the UV-B and UV-(A+B) findings as UV attenuation. The direct and
indirect effects of both UV-B enhancement and UV attenuation somewhat differed (Figs. 2a and b). Litter
decomposition increased when both the soil and plants were subjected to enhanced UV-B radiation (Fig.
2a), and UV-B enhancement significantly promoted N release when the plants were grown under UV-B
enhancement. Furthermore, P and lignin contents increased when the soil was under UV enhancement,
but neither the soil nor plants under UV-B enhancement affected C release. However, UV radiation
significantly affected litter decomposition when the soil was subjected to UV attenuation, and positive
litter decomposition effects were observed when plants were under UV attenuation (Fig. 2b). N release
showed the opposite effect between the soil and plants under UV attenuation.

### 3.3 Experimental conditions and plant types

A field experiment also showed increased and decreased of UV enhancement and attenuation on litter decomposition, respectively (Figs. 3a and b). Under laboratory conditions, the samples size was small, but based on the limited data, neither UV enhancement nor attenuation had an effect on litter decomposition and nutrient release, with the exception that weight loss occurred under UV attenuation.

Litter decomposition and N and P release from litter in the field experiment were significantly affected by changes in UV radiation. UV changes also affected litter decomposition differently for different litter types (Figs. 3c and d), but only the rate of decomposition, not the direction, was affected.

### 3.4 Experimental duration

The weight loss and nutrient release exhibited different RRs with incubation time (Fig. 4). UV

enhancement had no effect on the weight loss in the first four months of the experiment but did promote litter decomposition within 4 months to 18 months (Fig. 4a). As the incubation time progressed, UV enhancement had no effect on litter C release (Fig. 4b) and promoted the release of N in 6 months to 12 months, but showed N enrichment of litter after 18 months decomposition (Fig. 4c), and promoted the release of both P and lignin after 18 months (Figs. 4d and e). The UV attenuation negatively affected

weight loss, but the effect decreased as the incubation time increased (Fig. 4f). The C, N, P and lignin remaining presented different RRs in both direction and rate under UV attenuation (Figs. 4g-j).

### 3.5 Precipitation

Precipitation significantly affected litter decomposition rate under control treatment (Fig. 5a), especially in regions with relatively low precipitation, and the k decay increased as precipitation

increased. In addition, the effects of UV attenuation differed in different precipitation regions (Fig. 5b); compared with those in areas with relatively high precipitation, the effects of UV radiation in areas with relatively low precipitation were significantly lower.

### 3.6 Relationships between litter weight loss and nutrient release

Different effects of changes in UV radiation on the RR of remaining nutrients and weight remaining

were found (Fig. 6). For example, the RRs of remaining C (slope = 1.31) and N (slope = 1.23) were larger than the RR of the weight remaining under UV attenuation, which indicated that litter decomposition was more sensitive than was C and N release under UV attenuation conditions (Figs. 6a and b).



Interestingly, the RRs of remaining P and lignin changed to a relatively high degree during the initial stages but then decreased as the decomposition progressed under UV attenuation conditions (Figs. 6c and d), although the remaining P and lignin were more sensitive to UV enhancement.

### 3    Discussion

5      Litter decomposition is recognized as a complex process regulated by both biotic and abiotic factors. However, less recognized is how litter type (herb or wood), experimental conditions (field or laboratory), duration, and direct or indirect effects of UV exposure affect litter decomposition and nutrient cycling under UV exposure (Fig. S2) (Song et al., 2013a). In the present study, a meta-analysis was used to assess the effects of UV exposure on the dynamics of litter decomposition and nutrient release.

### 4.1 Differential responses of litter weight loss and nutrient release

The effects of changes in UV radiation on litter decomposition among different studies were contradictory. For example, UV enhancement had a positive effect on litter decomposition (Newsham et al., 2001) or no effect (Kirschbaum et al., 2011;Uselman et al., 2011;Song et al., 2014a), and it even led to a decrease (Pancotto et al., 2003;Newsham et al., 1997). Across all studies, the RR of the k decay was significantly correlated with changes in UV radiation (Fig. S3), which increased as the UV enhancement increased; furthermore, the RR of the k decay decreased as the UV attenuation increased but increased when the UV attenuation was lower than the threshold, indicating different sensitivities of litter decomposition to changes in UV radiation and a shift from direct to indirect effects of UV radiation (Song et al., 2013a). In addition, in the present study, the UV enhancement and attenuation had opposite effects on the litter decomposition and N and P release (Fig. 1). The UV enhancement positively affected the litter decomposition, but the UV attenuation reduced the litter decomposition; this reduction may have occurred because of enhanced photodecomposition or impacts to the abundance and community composition of microbial decomposers (Austin and Vivanco, 2006;Smith et al., 2010;Song et al., 2014b), which is consistent with the results of previous studies (Almagro et al., 2017;Gehrke et al., 1995;Song et al., 2013b;Pancotto et al., 2003). However, interestingly, changes in UV radiation did not affect the release of C, which was the focus of our concern, indicating that a different regulatory mechanism may be controlling litter decomposition.



The decomposition process can be affected by UV radiation received either by plants during growth or by litter during decomposition, which is due to differences in regulatory mechanisms. The direction and rate of litter decomposition and nutrient release also differed due to the direct and indirect effects of changes in UV radiation (Fig. 2), as the direct effects were regulated mainly by the effects of the UV radiation on the soil microorganisms, whereas the indirect effects were regulated mainly by the change in litter chemical properties (Pancotto et al., 2003). Under UV enhancement, the litter decomposition and nutrient release increased, and compared with that under no UV enhancement, the N release when plants were grown under UV enhancement was greater, which was mainly due to the higher initial litter N content (Figs. S4 and S5). These findings are consistent with those in the work of Song et al. (2013b)), who reported that UV enhancement increased the N content in litter and promoted litter decomposition. However, the N release was not affected when litter under UV enhancement during decomposition, which resulted from the balance between increased photodecomposition and decreased activity of decomposer organisms (Pancotto et al., 2003). In addition, under UV attenuation, the effect was greater on litter decomposition than on nutrient release, and the UV attenuation on the soil also reduced the litter decomposition, which was mainly due to a reduced effect of photodecomposition (Pancotto et al., 2005;Song et al., 2013a). However, UV attenuation exhibited no effect on the loss of litter biomass but did promote the release of N, which needs further investigation.

The effects of changes in UV radiation on litter decomposition also depend on the experimental conditions and litter type (Fig. 3). With the exception of the loss of biomass under UV attenuation, changes in UV radiation did not affect the litter decomposition or nutrient release; however, a significant effect was observed in the field studies, indicating differences between laboratory and field experiments. These results are consistent with those of previous laboratory-based studies (Kirschbaum et al., 2011;Lambie et al., 2014) in which limited effects of UV radiation may have been due to differences in moisture status between laboratory and field conditions. In addition, as expected, N and P release increased and decreased under UV enhancement and attenuation, respectively, but C release was not affected. Litter type affected only the rate of litter decomposition and not the direction in response to changes in UV radiation; these differences were due to the litter quality of the species (Day et al., 2015).

Litter decomposition is a temporal dynamic process, and the sensitivity of litter decomposition to changes in UV radiation depends on the incubation time (Wang et al., 2017). Thus, understanding the temporal dynamics of the impacts of UV radiation will greatly increase the predictability of litter



decomposition models within ecosystems. The duration of the experiments often produced inconsistent results. For example, neutral or even negative responses of litter decomposition to UV exposure were mostly recorded in short-term experiments (Kirschbaum et al., 2011;Lambie et al., 2014), whereas positive responses were often recorded in long-term experiments (Austin and Vivanco, 2006;Brandt et

al., 2010). In the present study, the litter decomposition also differed with decomposition time under UV enhancement and attenuation; the decomposition under both conditions exhibited three-stage temporal dynamics (Fig. 4). UV enhancement did not impact the weight loss or nutrient release during the early stage (0-4 months) but did significantly promote litter decomposition during the intermediate stage (4-18 months), indicating that the UV enhancement could accelerate litter decomposition, which requires a

sufficient period of UV accumulation (Wang et al., 2017), as well as nutrient release. However, UV attenuation significantly reduced the litter decomposition during the early stage, and the effect diminished as the decomposition time increased.

**4.2 Sensitivity of litter decomposition under UV change with precipitation**

In general, climatic (precipitation, temperature) and litter composition variables (C:N or lignin:N) are often used to predict weight loss rates (Gallo et al., 2009) because the activity of microorganisms that

decompose litter is regulated mainly by these variables. However, increasing numbers of studies have proven the key role of photodecomposition in arid ecosystems, which supplement biotic activity (Austin and Vivanco, 2006;Day et al., 2015;Gallo et al., 2009). In the present study, the k decay was still large in the areas of relatively low precipitation (Fig. 5), indicating that photodecomposition is an important factor

in arid ecosystems (Gallo et al., 2009;Almagro et al., 2017;Day et al., 2015), and the attenuation of radiation can even reduce decomposition by 60% (Austin and Vivanco, 2006). The k decay increased as the precipitation decreased from 866 mm to 110 mm, and previous studies have proven the positive roles of photodecomposition in litter decomposition in areas where the annual precipitation ranges from 152–726 mm (Huang et al., 2017). In addition, the RR of the k decay also indicated that UV attenuation

reduces litter decomposition, and the effect of UV attenuation on litter decomposition was greater under relatively low precipitation conditions than under relatively high precipitation conditions, demonstrating that the effects of UV radiation on litter decomposition differed under different climatic conditions (Ballare et al., 2011). This finding is consistent with the results of Brandt et al. (2007)) and Brandt et al. (2010)), who reported that the attenuation of UV had a greater effect on litter decomposition under

relatively dry experimental conditions than under relatively wet conditions, mainly because photodecomposition was the dominant driver of litter decomposition.

### 4.3 Relationship between weight loss and nutrient release under UV change

Across all studies, the weight loss and nutrient release were significantly correlated but were different under UV enhancement or attenuation (Fig. 6). The changes in UV radiation also affected the relationship between weight loss and nutrient release. The RRs of the remaining C and N were significantly correlated with the RR of weight remaining under UV attenuation, and the slopes were greater than 1. These findings indicated that the weight loss was more sensitive than nutrient release under UV attenuation, but these correlations were not observed under UV enhancement. However, the remaining lignin was negatively correlated with the weight remaining under UV attenuation, indicating that, compared with the decomposition of litter weight, the release of lignin occurred more quickly at the beginning of decomposition but became enriched during the later stages of decomposition (McClaugherty and Berg, 1987). Together, these results indicated that there were differences in the effects of UV radiation on the loss of litter weight and nutrient release, which should be investigated in future studies.

### 4.4 Implications and uncertainties

Our study revealed that responses of nutrient release and weight loss in litter decomposition to UV change differed, and the responses depended on the direct or indirect effect of UV, experimental conditions and experiment duration. The results have clarified how UV changes affected weight loss and nutrient release in litter decomposition. However, there are several critical challenges for further studies. First, most studies have concentrated on the effect of UV on litter weight loss, but less attention has been paid to the release and ultimate fate of litter C and N (storage in soil or emission to the atmosphere), which were the focus of concern here (Wang et al., 2015).

Second, climate change usually involves changes in multiple environmental factors, such as rising temperature and changes in precipitation patterns, which might simultaneously and interactively alter the litter decomposition (Brandt et al., 2007;Zepp et al., 2007), but few studies focused photodecomposition in the context of global changes. Therefore, to better understand the effects of UV in a changing global environment, further studies should investigate how climate change affects the balance between positive

and negative effects on litter decomposition in response to changes in UV.

## 4 Conclusions

The current synthesis study, which was based on global data, shows that UV radiation plays an important role in litter decomposition and that the effect of photodecomposition becomes stronger as
precipitation decreases below 855 mm, at which point litter decomposition becomes sensitive to UV attenuation. The effects on litter decomposition also depend on the direct and indirect effects of changes in UV radiation, as UV attenuation on plants does not affect the litter weight loss. In addition, with the exception of the loss of litter weight under conditions of UV attenuation, changes in UV radiation under laboratory conditions do not affect litter decomposition or nutrient release, and the weight loss under UV
radiation is characterized by three-stage temporal dynamics. Furthermore, the relationship between weight loss and nutrient release is altered under UV attenuation, as C and N release occur more slowly than the weight loss, suggesting that, in the future, studies should consider nutrient release instead of focusing mainly on the effects of UV radiation on litter weight loss.

*Author contributions.* W. Y conceived and led the study. All authors contributed to writing and editing the manuscript.

*Competing interests.* The authors declare that they have no conflict of interest.

*Acknowledgments.* The study was funded by the National Postdoctoral Program for Innovative Talents (BX201700200), the China Postdoctoral Science Foundation (2018M631200), the Fundamental Research Funds for the Central Universities (2452017233), the National Key Research and Development Program of China (2016YFC0501605) and the National Natural Science Foundation of China (41390463, 31370425). The authors declare that they have no competing interests.

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





**Figure legends**

**Figure 1** Effects of UV treatment on litter decomposition and remaining nutrients. The black symbols

indicate significant differences (p < 0.05) between the response ratios (RRs) and zero. The vertical dotted

line represents a mean effect size of 0. The sample size for each variable is shown and represents the UV-

B enhancement and attenuation and UV-(A+B) attenuation from left to right.

**Figure 2** The direct (litter decay directly exposed to UV) and indirect (plants received UV during growth)

effects of UV enhancement (a) and attenuation (UV-B and UV-(A+B)) (b) on litter decomposition and

remaining nutrients. The black symbols indicate significant differences (p < 0.05) between the response

ratios (RRs) and zero. The sample size for each variable is shown and represents the indirect and direct

effect from left to right.

**Figure 3** Effects of UV enhancement (a) and attenuation (UV-B and UV-(A+B)) (b) on litter

decomposition and remaining nutrients in laboratory and field experiments as well as the effects of UV

enhancement (c) and attenuation (d) on litter decomposition and remaining nutrients in herb and wood

litters. The black symbols indicate significant differences (p < 0.05) between the response ratios (RRs)

and zero. The sample size for each variable is shown and represents the laboratory and field experiments

(a and b, respectively) as well as the herb and wood litters (c and d, respectively), from left to right.

**Figure 4** Effects of UV enhancement (a-e) and attenuation (UV-B and UV-(A+B)) (f-j) on litter

decomposition and remaining nutrients during decomposition. The dashed line supposed to represent

zero, above the dashed line indicate retarded litter degradation and values below the line indicate

enhanced litter degradation. The black symbols indicate significant differences (p < 0.05) between the

response ratios (RRs) and zero. The sample size for each variable is shown.

**Figure 5** Relationship between the k decay and mean annual precipitation (a) and relationships between

response ratios (RRs) of the k decay and mean annual precipitation under UV attenuation (UV-B and

UV-(A+B)) (b).

**Figure 6** Relationships of the response ratios (RRs) of the remaining nutrients and litter weight remaining.

The values of nutrients and weight remaining above zero mean decomposition was retarded by UV,

whereas below zero mean decomposition was enhanced by UV. The slope value is based on all data, p

values between remaining nutrients and litter weight are shown under UV enhancement (UV+) and

attenuation (UV-B and UV-(A+B)) (UV-), and black dashed lines represent the regression equations that

show significant correlations between remaining nutrients and litter weight remaining either under UV+





or UV- conditions.





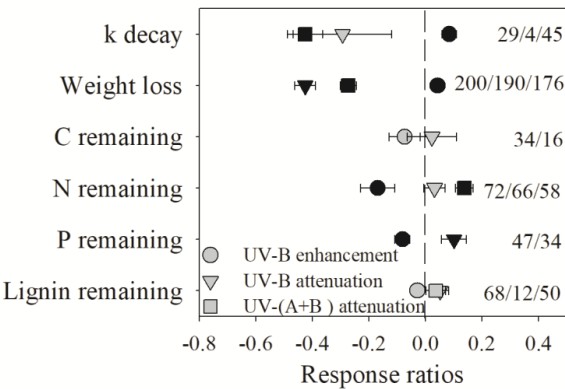

**Figure 1**





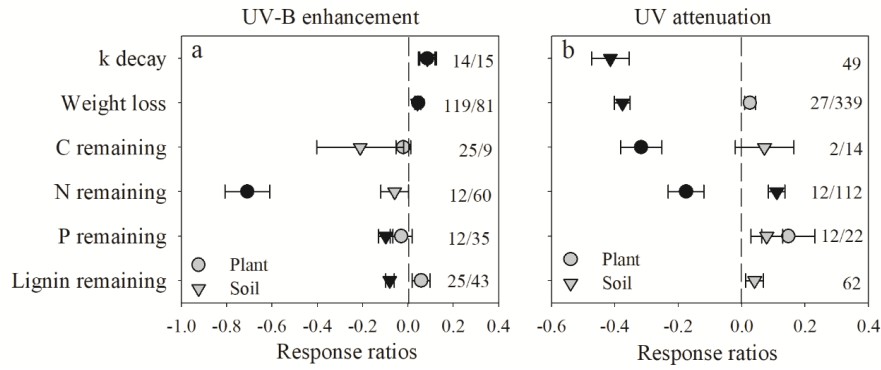

**Figure 2**





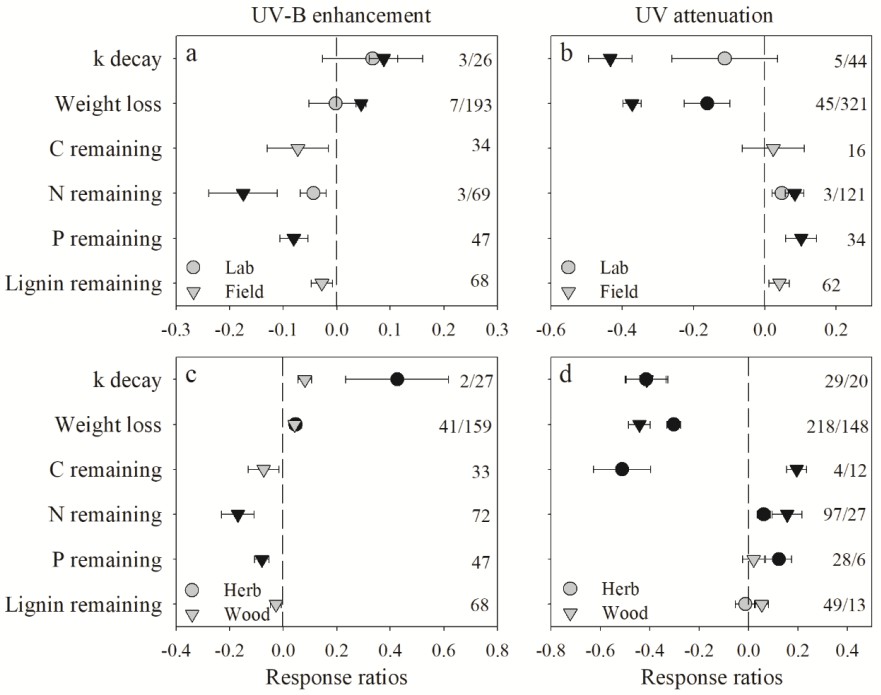

**Figure 3**



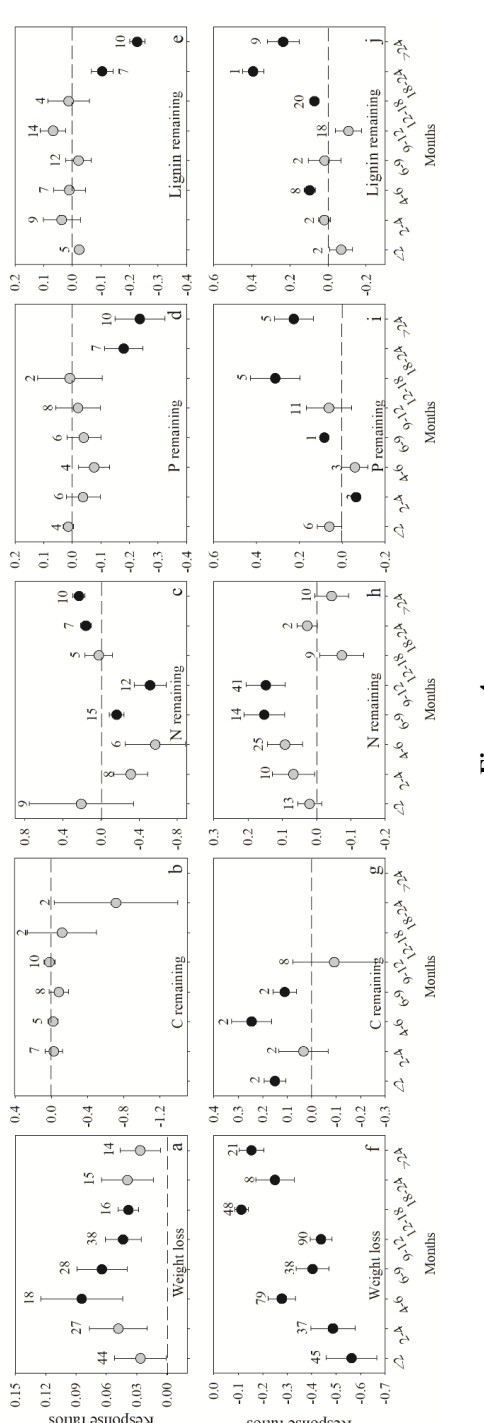

**Figure 4**





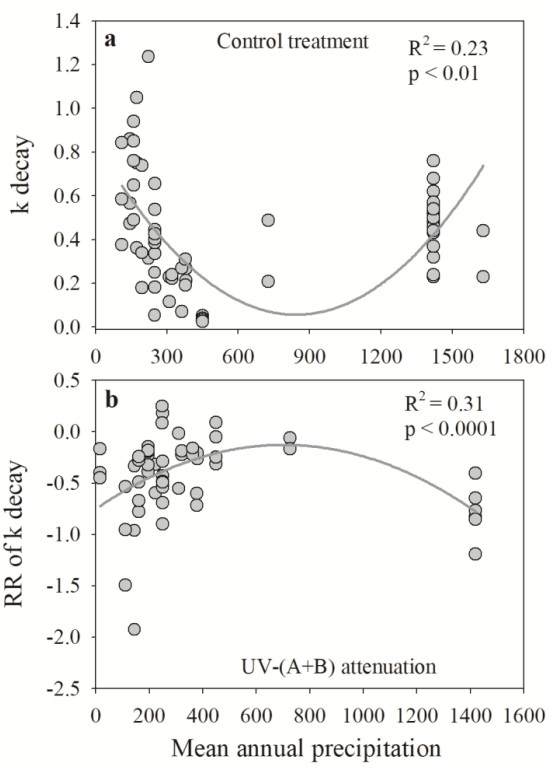

**Figure 5**



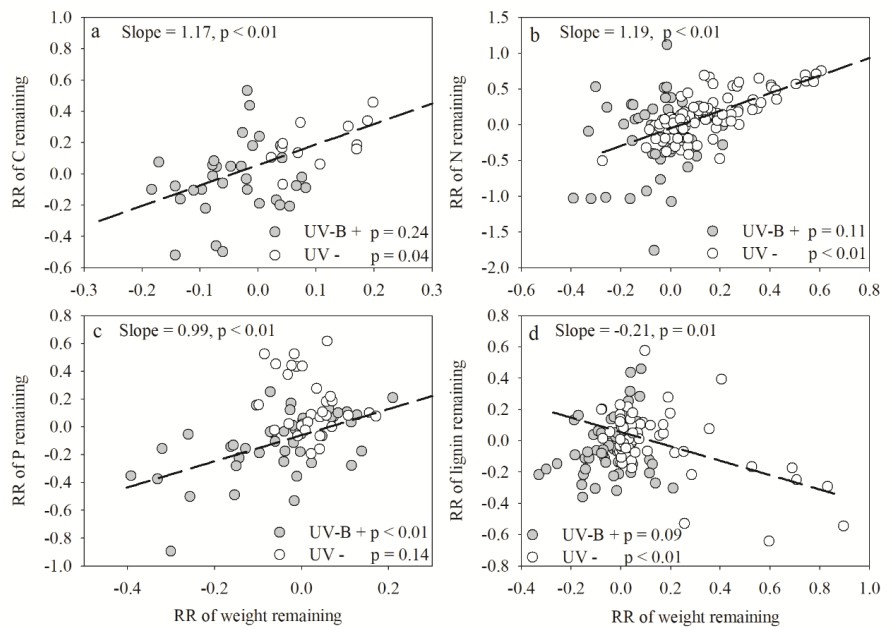

**Figure 6**