# Peer review of "Different sensitivities of litter decomposition and nutrient release to ultraviolet radiation"

_Biogeosciences, 2018_

## Referee Comment (RC1) · Anonymous Referee #1 · 13 Nov 2018

bg-2018-445 comments

I have read with interest the manuscript of Yan and co-authors titled "Different sensitivities of litter decomposition and nutrient release to ultraviolet radiation". This meta-analysis synthesized global data of UV changes on litter mass loss, C, N and P release and lignin degradation during decomposition. The authors also discussed the magnitudes of UV treatments on litter decomposition among various experimental conditions, litter types, durations and precipitation levels. In general, I found this paper to not be well written, especially the discussion. I also found some terminologies and presentations to be confusing in some cases. But I think the data are scientifically valid and interesting to the wider community, so I think this manuscript is suitable for publication in Biogeosciences after reversion. Here, I provide some comments which the authors

may find useful.

General comments: A large part of the results were presented again in the Discussion. A rewriting of the results should allow to extract more explicitly the meaning of the findings avoiding the need to be repeated in the discussion. The authors have made many classifications, i.e., lab/field, litter types (although not very correct and could be re-divided), durations and MAP levels, but some interesting findings were not presented. For example, why UV enhancement and attenuation had opposite effects on mass loss vs. N/P release (Fig. 1)? Why UV changes had strong effects on litter N content and C/N and lignin/N ratios (Fig. S4)? The authors made a very simple regression result in Fig. S5, which, however, I think cannot help us to answer the above two questions. Instead, for the Fig. S5, why not to try to separate the control and treatment data because UV treatment did had very strong influence on litter N content and associated ratios, and this may help us to make insightful discussion. I remember that the photo-degradation of litter decomposition was found in arid grassland (i.e., Austin 2006 Nature). Yes, the authors compared the decomposition rate and MAP, and from that figure, the regression results were interesting, particularly in arid areas with low precipitation. However, it was much different when the MAP reached at 1400 mm, so I think the regression results can be presented as bars with different categories (i.e., MAP ranging from 0-50 mm, 50-100 mm, etc.). Of course, the previous Figure 5 can be presented in Supplementary Information. Following the above question, photo-degradation of litter decomposition may be observed in grasslands in arid ecosystems in previous years, but in recent years, there were many studies conducted in forests. Therefore, why not divide the litter type to more specific classifications (i.e., grass, herb, broad-leaved and needle foliage) corresponding to the ecosystem types (grasslands vs. forests)? By the way, I think just use "herb" in grasslands was not correct and the authors should carefully distinguish grass and herb.

Detailed comments: Line 30. "weight loss"? And why not "mass loss"? The latter one is more widely used in litter decomposition studies. Line 222. The authors

declare that "UV-(A+B) attenuation. . . but showed LITTLE effect. . ."; however, "UV-B enhancement. . . showed SIGNIFICANT effect" (line 221). I think the authors should change the presentation. In fact, the RR for UV-(A+B) attenuation was less than -0.25, but that for UV-B enhancement was only 0.04, so why did the authors say UV-B enhancement had SIGNIFICANT effect whereas UV-(A+B) attenuation has LITTLE effect? In fact, the RR for UV-B enhancement maybe not significant (overlap with zero) if the sample size was smaller. Lines 223-225. The authors declared that the RR was greater for k decay compared with mass loss, but I think the authors should treat the k decay and mass loss results with caution. There are at least two reasons: First, both the k decay (assumed that the exponential models were used) and mass loss reflect one thing. Second, the sample size for UV-B attenuation on k decay was very small (n=4), so its confidence was not strong, and this result may be excluded in some cases. Therefore, I think using other sentences (i.e., Similarly, k decay. . .) would be better than ". . . greater than . . .". Line 225. Why the authors did not present some results on N and P release directly? That will be very interesting because it seems that UV-B enhancement and attenuation showed opposite effects on N/P release relative to on mass loss/k rate. And then C and lignin. Line 225. I think "no effect" was not a very good word if we have other choice. Why not "the effects were not significant"? Line 257. "UV enhancement had NO effect on the weight loss in the first four months"? I am confused why some RRs were not significant (marked in gray) but the 95% CI did not overlap with zero. This problem can be found in many figures. Please check it. Figure 267-268. What is "control treatment"? "Control + treatment"? From Figure 1, UV treatment had very strong influence on k rate, so have you tried to compare the relationships between MAP and k rate under control and treatment conditions separately? Just like Figure 6. Lines 275-278. I am confused that how can the authors conclude that . . . was more sensitive than . . .. The slopes >1? Or compared with the 1:1 lines in Figure 6? Lines 284-289. It seems that this sentence was redundant here because this has been presented in the Introduction. We should focus on the most important findings, and some general sentences, i.e., litter decomposition is a complex process regulated

by both biotic and abiotic factors, were not very interesting for readers. Lines 292-295. Just like the suggestion mentioned above, the sentence "UV enhancement had . . . lead to a decrease" has been introduced in the Introduction section, so we do not need to repeat it again here. Line 302. I think "litter decomposition" should be replaced by "mass loss" here. As you declared (but I suggest to delete it) that "litter decomposition is a complex process" and this process includes many sub-process, i.e., mass loss we observed, C structure breakdown, release of N, P and other nutrients, etc. Therefore, UV treatments had opposite effects on mass loss vs. N/P release and not litter decomposition vs. N/P release. Line 310. I think the authors should discuss the sample size for remaining C, and not only "a different regulatory mechanism". In meta-analysis, the collected data have strong influence on our conclusion, which may be biased if the sample size is too small. This is a potential uncertainty in meta-analysis.

Figures. The authors should note that what are the meanings of the error bars, e.g., 95% CI. Figure 1. Maybe there was a small mistake for the UV-B attenuation (triangle) for "k decay" in Figure 1. Generally, the difference can be considered as significant if the 95% CI did not overlap with zero. However, for the UV-B attenuation for "k decay", the 95% CI did not overlap with zero, but it was gray and not black, although the sample size was very small (n=4).

---

## Referee Comment (RC2) · Anonymous Referee #2 · 26 Nov 2018

Yan et al. conducted a meta-analysis to compare the sensitivities of litter mass loss and nutrient release to UV manipulation. This study has the potential to contribute to the current literature on the roles of UV radiation in terrestrial biogeochemical cycles. However, there are significant issues with the presentation of results and the use of English language. Several figures are very hard to understand because of poor figure captions and missing methodological details. As there have been three meta-analyses on the same general topic of the UV effects on decomposition processes, a significant portion of the results did not offer much new information. The UV effects on litter mass loss and nutrient release were not compared in a statistically meaningful way.

Specific comments: Abstract, the abstract does not include any result regarding the comparison of UV effects on litter mass loss and nutrient release, which was supposed

to be the main research question, judging from the title.

Page 1 Lines 20, 25, 'sensitivity' is a vague term. Please define how sensitivity was quantified.

P1L22-24, I cannot follow this sentence. If the key is 'three-stage pattern', then describe what these three stages are.

P1L25, the final sentence was too generic. It's not clear how this study helps to move the field forward.

Introduction, in general, I find the introduction quite wordy and lack of focus. For example, the authors discussed the specific UV effects on decomposition twice in two paragraphs. Some knowledge gaps that the authors identified have been studied by previous meta-analysis, e.g., the effects of precipitation (P3L14-15) and the effects of experimental duration (P4L26). Please think carefully about the unique contributions of this study and highlight them in the introduction.

P2L7, UV radiation doesn't directly contribute to photosynthesis or C fixation.

P2L27, this is a good place to discuss the indirect effects of UV radiation.

P3L4, the term 'photodecomposition' needs to be defined.

P3L4, try 'sparse vegetation'

P3L19-30, this is the 2nd time the authors described the specific pathways through which UV radiation affected decomposition processes (1st time: P2L23-27). Please try pooling these materials together.

P3L29, explain 'the effects of UV enhancement _on the soil_'.

P4L10-24, this paragraph introduces the main novelty of this study: the UV effects on C vs. nutrient release. Consider highlighting this instead of burying it at the end of introduction.

P4L13, King et al. 2012 Biogeochemistry also included a meta-analysis.

P4L18, compared to previous meta-analyses, this study included a smaller number of published studies (i.e., 54) due to the goal of comparing C vs. nutrient release. I don't find it fair to criticize the sample sizes of previous studies.

P4L27, I don't understand how the authors defined 'datasets'. A paired observation or a published experiment?

P5L19, what are the chemical properties included in this study? Please list all of them here. This is a good place to explain how 'C/N/P remaining' or 'C/N/P release' were calculated. In addition, does 'wood' mean woody tissues or leaves of woody species?

Results, direct and indirect effects of UV radiation represent fundamentally different processes. One involves exposing litter to various levels of UV radiation, while the other refers to growing plants under UV manipulation and then accessing decay processes. Figure 2 is the only place where the direct and indirect were differentiated, but I find it impossible to understand. I have the impression that the direct and indirect effects were mixed together in most figures, except in Figure 2. I strongly believe that mixing the two is wrong.

In all figures, it's not clear how sample size was counted. Does it include the number of replicates in original studies?

Figure 1, are the results about the 'direct effects' only? What does the error bar represent? It is more intuitive to show 95% confidence intervals with error bars.

Figure 1, I find it very surprising that there were less published studies on C remaining than on P remaining.

Figure 2, I am having a hard time understanding this figure. Were soil and plant supposed to represent direct and indirect effects, respectively?

P7L16, please consider merging UV-B and UV-(A+B) results in the figure 1.

P8L1, you mean 'results from field experiments'?

Figure 4, if one experiment lasted exactly 4 months, was it categorized as '2-4' or '4-6' months? Please add text in the figure to cue readers about UV amendment vs. attenuation.

Figure 4, there are not enough data over 500 mm precipitation to draw reliable conclusion.

P8L25-26, use of stats is questionable. The authors need to compare the slopes with 1 statistically.

P9L5-9, a summary of key results will be more useful to start the discussion.

The discussion overall does not offer much original interpretation of data. In many places, it repeats the results or cites the findings from literature (e.g., P9L20-25, P9L6-10).

P9L14-15, this should be presented in the results.

P10L20, you mean no effects were observed in the lab?

P11L7, Figure 4 presents the data in an interesting way. I consider it a stronger component of the manuscript. However, it is debatable whether the temporal patterns clearly show three different stages. For example, the UV attenuation results seem to follow a two-stage pattern.

---

## Author Comment (AC1) · 8 Jan 2019

Responses and corresponding revisions:
General comments: 1: A large part of the results were presented again in the Discussion. A rewriting of the results should allow to extract more explicitly the meaning

of the findings avoiding the need to be repeated in the discussion. The authors have made many classifications, i.e., lab/field, litter types (although not very correct and could be re-divided), durations and MAP levels, but some interesting findings were not presented. For example, why UV enhancement and attenuation had opposite effects on mass loss vs. N/P release (Fig. 1)? Why UV changes had strong effects on litter N content and C/N and lignin/N ratios (Fig. S4)? Response: Thank you for your comments. Follow your suggestion, we have rewritten the results section to more explicitly describe the findings and have revised other sections accordingly. Fig. 1 shows that UV change had effects on mass loss opposite those of N/P remaining, which indicates that mass loss exhibited a similar change as did N/P release. The concentration of N/P may increase over the course litter decomposition and result in a negative value of N/P release, which cannot be used in the calculation of the formula; thus, we used N/P remaining in the calculation. Potential reasons for the large differences among the effects of UV on leaf litter chemistry in the study include the following: 1) interspecific differences, as the magnitude of UV effects on leaf litter quality varies with plant species and type, for example, between herbaceous plants (Pancotto et al., 2003; Pancotto et al., 2005) and woody plants (Song et al., 2013b); 2) variation in environmental factors, especially precipitation, temperature, and soil nutrient content, which influence litter chemistry (the studies were performed at different sites, including dune grassland (Hoorens et al., 2004) and field sites (Pancotto et al., 2003; Pancotto et al., 2005)); and 3) variation in treatment duration, because the accumulation of variation in chemistry parameters changes with exposure time. The length of time that plants were grown under UV radiation ranged from one growth season to 3 years in the included studies. These reasons might explain the strong effects on litter chemistry.

2: The authors made a very simple regression result in Fig. S5, which, however, I think cannot help us to answer the above two questions. Instead, for the Fig. S5, why not to try to separate the control and treatment data because UV treatment did had very strong influence on litter N content and associated ratios, and this may help us to make insightful discussion. Response: Thank you for your comments. Follow your suggestion, we separated the control and treatment data in the revised manuscript (Fig. S8). The results indicated that k decay had a significant correlation with N concentration under both the control and UV-change treatments. Furthermore, the slope of the relationship between k decay and N concentration was larger under UV enhancement than under ambient environment conditions and UV attenuation. We have added this information to the revised manuscript.

3: I remember that the photo-degradation of litter decomposition was found in arid grassland (i.e., Austin 2006 Nature). Yes, the authors compared the decomposition rate and MAP, and from that figure, the regression results were interesting, particularly in arid areas with low precipitation. However, it was much different when the MAP reached at 1400 mm, so I think the regression results can be presented as bars with different categories (i.e., MAP ranging from 0-50 mm, 50-100 mm, etc.). Of course, the previous Figure 5 can be presented in Supplementary Information. Response: Thank you for your suggestion. According to your suggestion, we have presented the results as bars rather than in a regression plot and have added the comparison of k decay between the control and UV-attenuation treatments. The results showed the UV attenuation had a significant effect at precipitation levels ranging from 100 to 200 mm and from 1400 to 1500 mm. Moreover, we moved the original Fig. 5 to Supplementary Information (Fig. S6).

4: Following the above question, photo-degradation of litter decomposition may be observed in grasslands in arid ecosystems in previous years, but in recent years, there were many studies conducted in forests. Therefore, why not divide the litter type to more specific classifications (i.e., grass, herb, broad-leaved and needle foliage) corresponding to the ecosystem types (grasslands vs. forests)? By the way, I think just use "herb" in grasslands was not correct and the authors should carefully distinguish grass and herb. Response: Thank you for your comment. According to your suggestion, we have divided the litter types according to ecosystem and have added the results for specific groups (i.e., grasses, herbs, and broad-leaved and needle-leaved plants) in

Supplementary Information (Fig. S5). In addition, associated content has been added to the revised manuscript.

Detailed comments 5. Line 30. "weight loss"? And why not "mass loss"? The latter one is more widely used in litter decomposition studies. Response: Thank you for your suggestion. We have replaced 'weight loss' with 'mass loss' in the manuscript and figures.

6. Line 222. The authors declare that "UV-(A+B) attenuation: : : but showed LITTLE effect: : :"; however, "UV-B enhancement: : : showed SIGNIFICANT effect" (line 221). I think the authors should change the presentation. In fact, the RR for UV-(A+B) attenuation was less than - 0.25, but that for UV-B enhancement was only 0.04, so why did the authors say UV-B enhancement had SIGNIFICANT effect whereas UV-(A+B) attenuation has LITTLE effect? In fact, the RR for UV-B enhancement maybe not significant (overlap with zero) if the sample size was smaller. Response: Thank you for your comment. We have merged the UV-B and UV-(A+B) results into the category UV attenuation in Fig. 1 as proposed by Reviewer #2, and we have moved the original UV-B and UV-(A+B) results to the supporting information (Fig. S2). The text has been revised as follows: 'As expected, UV enhancement and attenuation showed opposite effects on mass loss and nutrient release. UV enhancement and attenuation showed significant effects on k decay, with RRs of 0.09 and -0.41 (Fig. 1), respectively; furthermore, UV-B enhancement and attenuation showed significant effects on mass loss, with RRs of 0.04 and -0.35, respectively. UV enhancement promoted N and phosphorus (P) release, with RRs of -0.16 and -0.08 of N and P remaining, respectively. UV attenuation showed the opposite effects on N and P remaining, with RRs of 0.08 and 0.10. The effects of changes in UV radiation on C and lignin release were not significant. Both UV-(A+B) and UV-B attenuation showed similar effects on mass loss and N and P release (Fig. S2)'.

7. Lines 223-225. The authors declared that the RR was greater for k decay compared with mass loss, but I think the authors should treat the k decay and mass loss results

with caution. There are at least two reasons: First, both the k decay (assumed that the exponential models were used) and mass loss reflect one thing. Second, the sample size for UV-B attenuation on k decay was very small (n=4), so its confidence was not strong, and this result may be excluded in some cases. Therefore, I think using other sentences (i.e., Similarly, k decay: : :) would be better than ": : : greater than : : :". Response: Thank you for your comment. We agree with your statement that both k decay and mass loss reflect one thing. Following your suggestion, we have revised the sentence to ' UV enhancement and attenuation showed significant effects on k decay, with RRs of 0.09 and -0.41 (Fig. 1), respectively; furthermore, UV-B enhancement and attenuation showed significant effects on mass loss, with RRs of 0.04 and -0.35, respectively.'

8 Line 225. Why the authors did not present some results on N and P release directly? That will be very interesting because it seems that UV-B enhancement and attenuation showed opposite effects on N/P release relative to on mass loss/k rate. And then C and lignin. Response: Thank you for your comment. Fig. 1 shows that UV change had effects on mass loss opposite those of N/P remaining, which indicates that mass loss exhibited a similar change as did N/P release. We originally planned to show the N/P release results. However, the concentration of N/P may increase over the course litter decomposition and result in a negative value of N/P release, which cannot be used in the calculation of the formula; thus, we used N/P remaining in the calculation.

9. Line 225. I think "no effect" was not a very good word if we have other choice. Why not "the effects were not significant"? Response: Thank you for your suggestion. We have revised this sentence to 'The effects of changes in UV radiation on C and lignin release were not significant' in the revised manuscript.

10. Line 257. "UV enhancement had NO effect on the weight loss in the first four months"? I am confused why some RRs were not significant (marked in gray) but the 95% CI did not overlap with zero. This problem can be found in many figures. Please check it. Response: Thank you for your comment. We apologize for the misunderstanding regarding the values in the figures. It is true that an RR was considered significant when its 95% confidence interval (CI) did not overlap with zero. The value in the figure indicates the weighted response ratio, and the error bar represents the standard error, not the 95% CI.

11. Figure 267-268. What is "control treatment"? "Control + treatment"? From Figure 1, UV treatment had very strong influence on k rate, so have you tried to compare the relationships between MAP and k rate under control and treatment conditions separately? Just like Figure 6. Response: Thank you for your comment. In the control treatment, litter decomposition was occurring under ambient environment conditions, without the enhancement or attenuation of UV. Our aims were to explore the relationship between k decay and MAP under ambient conditions and determine whether the effect of UV attenuation on k decay is affected by MAP. In addition, the first figure displays the relationship between k decay and MAP, and the later one shows the relationship between the RR of decay and MAP; we cannot compare these two regression equations due to their different parameters.

12. Lines 275-278. I am confused that how can the authors conclude that : : : was more sensitive than : : :. The slopes >1? Or compared with the 1:1 lines in Figure 6? Response: Thank you for your comment. We apologize for the lack of clarity. We have revised the text to 'Various effects of changes in UV radiation on the RRs of remaining nutrients and weight remaining were found (Fig. 6). The slope of the RRs of remaining C and N and the weight remaining under UV attenuation were 1.31 and 1.23, respectively, however, the effects of both UV enhancement and UV attenuation on the relationships between each of C, N and P and mass loss relative to the ambient environment were not significant (p>0.05). Interestingly, UV enhancement significantly promoted lignin release compared with mass loss (p<0.01).'

13. Lines 284-289. It seems that this sentence was redundant here because this has been presented in the Introduction. We should focus on the most important findings, and some general sentences, i.e., litter decomposition is a complex process regulated

by both biotic and abiotic factors, were not very interesting for readers. Response: Thank you for your comment. We agree with your statement that the sentence was redundant and have deleted it from the revised manuscript. In response to this comment and a suggestion proposed by Reviewer 2, we have revised the beginning of the discussion section to 'In the present study, a meta-analysis was performed to assess the effects of UV exposure on the dynamics of litter decomposition and nutrient release. We found that leaf source (grassland or forest), experimental condition (field or laboratory), experimental duration, and direct or indirect effects of UV exposure affected litter decomposition and nutrient release under UV exposure'.

14. Lines 292-295. Just like the suggestion mentioned above, the sentence "UV enhancement had : : : lead to a decrease" has been introduced in the Introduction section, so we do not need to repeat it again here. Response: Thank you for your comment. We have deleted this sentence from the revised manuscript.

15. Line 302. I think "litter decomposition" should be replaced by "mass loss" here. As you declared (but I suggest to delete it) that "litter decomposition is a complex process" and this process includes many sub-process, i.e., mass loss we observed, C structure breakdown, release of N, P and other nutrients, etc. Therefore, UV treatments had opposite effects on mass loss vs. N/P release and not litter decomposition vs. N/P release. Response: Thank you for your comment. Following your suggestion, we have replaced 'litter decomposition' with 'mass loss.'

16. Line 310. I think the authors should discuss the sample size for remaining C, and not only "a different regulatory mechanism". In meta-analysis, the collected data have strong influence on our conclusion, which may be biased if the sample size is too small. This is a potential uncertainty in meta-analysis. Response: Thank you for your comment. We agree with your observation that the sample size may have affected the results of the meta-analysis. Thus, we have revised the text to 'However, interestingly, changes in UV radiation did not affect the release of C, which was a focus of our concern, may be indicated that a different regulatory mechanism other than

UV radiation may be controlling litter decomposition, although the small sample size may have contributed to the insignificant results. Therefore, more studies needed to determine the effects of UV changes on the release of C'.

17. Figures. The authors should note that what are the meanings of the error bars, e.g., 95% CI. Figure 1. Maybe there was a small mistake for the UV-B attenuation (triangle) for "k decay" in Figure 1. Generally, the difference can be considered as significant if the 95% CI did not overlap with zero. However, for the UV-B attenuation for "k decay", the 95% CI did not overlap with zero, but it was gray and not black, although the sample size was very small (n=4) Response: Thank you for your comment. We apologize for the misunderstanding regarding the values in the figures. The values in the figure are the weighted response ratios, and the error bars represent standard error. The black symbols indicate significant differences ($p < 0.05$) between the response ratios and zero. We have revised the figure legends for clarity.

Please also note the supplement to this comment:
https://www.biogeosciences-discuss.net/bg-2018-445/bg-2018-445-AC1-supplement.zip

---

## Author Comment (AC2) · 8 Jan 2019

1: Yan et al. conducted a meta-analysis to compare the sensitivities of litter mass loss and nutrient release to UV manipulation. This study has the potential to contribute to the current literature on the roles of UV radiation in terrestrial biogeochemical cycles. Response: Thank you for your detailed review and helpful suggestions on the manuscript. We have revised the manuscript according to your comments. Each comment is addressed below, and the revised manuscript can be found in the Supplement.

2: However, there are significant issues with the presentation of results and the use of English language. Several figures are very hard to understand because of poor figure captions and missing methodological details. As there have been three metaanalyses on the same general topic of the UV effects on decomposition processes, a significant portion of the results did not offer much new information. The UV effects on litter mass loss and nutrient release were not compared in a statistically meaningful way. Response: Thank you for your suggestions. We agree with your statement that there have been three meta-analyses on the same general topic of the UV effects on decomposition processes. One of these studies mainly emphasized the litter weight remaining and its chemistry under elevated UV radiation (Wang et al. 2015), and the others examined only the litter weight remaining under changes in UV-B radiation (King et al., 2012; Song et al., 2013a). In general, the loss of litter mass increases as decomposition time increases, but nutrient release may show a different pattern. For example, the nitrogen (N) remaining in litter was shown to increase after fifteen months of photodegradation of litter decomposition in semiarid Mediterranean grasslands (Almagro et al., 2017). Thus, to better understand the C and nutrient release from litter, clarification of the correlation between mass loss and nutrient release during litter decomposition under changes in UV radiation is urgently needed. To clarify the effects of UV radiation on litter decomposition, especially its effects on C and nutrient release during the litter decomposition process, we conducted this study. Our main goal was to resolve the conflicting results presented to date and to clarify the response of nutrient release to UV radiation, which may differ from that of the rate of litter mass loss. We have revised the relevant sections of the revised manuscript accordingly. In addition, we have added the results of the statistical analysis of nutrients release and mass loss under UV changes relative to ambient environment conditions, and we have carefully revised the figure captions. The paper has been edited for English language by American Journal Experts. We believe the quality of the manuscript as been markedly improved.

3: Abstract, the abstract does not include any result regarding the comparison of UV effects on litter mass loss and nutrient release, which was supposed to be the main research question, judging from the title. Response: Thank you for your comments. We apologize for the lack of information on UV effects on litter mass loss and nutrient

release. Following your suggestion, we have added the relevant results to the abstract. We did not compare the sensitivities of litter mass loss and nutrient release in this study; accordingly, we have revised the title to 'Responses of litter decomposition and nutrient release to ultraviolet radiation: a meta-analysis.'

4: Page 1 Lines 20, 25, 'sensitivity' is a vague term. Please define how sensitivity was quantified. Response: Thank you for your comments. Our use of the term sensitivity was in appropriate because we did not evaluate differences; thus, we have deleted this term from the manuscript.

5: P1L22-24, I cannot follow this sentence. If the key is 'three-stage pattern', then describe what these three stages are Response: Thank you for your comment. We have revised the text to 'In addition, mass loss and nutrients release under UV radiation varied over the decomposition process...'

6: P1L25, the final sentence was too generic. It's not clear how this study helps to move the field forward. Response: Thank you for your comments. We have revised the sentence to make it more specific as follows: 'Overall, changes in UV had considerable effects on both litter mass loss and nutrient release, suggesting that changes in UV radiation may greatly impact C and nutrient cycling in terrestrial ecosystems.'

7: Introduction, in general, I find the introduction quite wordy and lack of focus. For example, the authors discussed the specific UV effects on decomposition twice in two paragraphs. Some knowledge gaps that the authors identified have been studied by previous meta-analysis, e.g., the effects of precipitation (P3L14-15) and the effects of experimental duration (P4L26). Please think carefully about the unique contributions of this study and highlight them in the introduction Response: Thank you for your suggestion. Accordingly, we have deleted the redundant sentences and adjusted the structure to clarify the focus of the introduction section. For example, we merged the fourth paragraph with the second one. We agree with your comment that effects of precipitation (P3L14-15) and experimental duration (P4L26) on decomposition have been studied.

However, whether the effect of UV change on decomposition remains consistent over time and how litter decomposition varies over time remain unclear. We have revised the relevant sentences in the introduction section of the revised manuscript to include this information.

8: P2L7, UV radiation doesn't directly contribute to photosynthesis or C fixation. Response: Thank you for your comment. It is true that UV radiation does not directly contribute to photosynthesis or C fixation; thus, we deleted the sentence.

9. P2L27, this is a good place to discuss the indirect effects of UV radiation. Response: Thank you for your suggestion. Following your suggestion, we moved the content addressing the indirect effects of UV radiation to this section and have carefully revised the text.

10. P3L4, the term 'photodecomposition' needs to be defined. Response: Thank you for your comment. Photodegradation is the breakdown of organic matter via solar radiation. We have defined the term in the revised manuscript.

11. P3L4, try 'sparse vegetation' Response: Thank you for your suggestion. We have used 'sparse vegetation' instead of 'little amount of vegetation' in the manuscript.

12. P3L19-30, this is the 2nd time the authors described the specific pathways through which UV radiation affected decomposition processes (1st time: P2L23-27). Please try pooling these materials together Response: Thank you for your comment. Follow your suggestion, we have adjusted the structure and merged the content together.

13. P3L29, explain 'the effects of UV enhancement _on the soil_'. Response: Thank you for your comment. For clarity, we have revised the sentence to 'Experimental studies of the effects of UV enhancement on plants have shown increases, no change or decreases in litter decomposition (Newsham et al., 2001; Hoorens et al., 2004; Song et al., 2013b) as well as the experimental results of litter decomposition when the soil under UV enhancement (Moody et al., 2001; Gehrke et al., 1995).'

14. P4L10-24, this paragraph introduces the main novelty of this study: the UV effects on C vs. nutrient release. Consider highlighting this instead of burying it at the end of introduction. Response: Thank you for your comment. We have revised the introduction to highlight the novelty of this study.

15. P4L13, King et al. 2012 Biogeochemistry also included a meta-analysis. Response: Thank you for your comment. We have read the study carefully and have added relevant information to the manuscript.

16. P4L18, compared to previous meta-analyses, this study included a smaller number of published studies (i.e., 54) due to the goal of comparing C vs. nutrient release. I don't find it fair to criticize the sample sizes of previous studies Response: Thank you for your comment. We apologize for the inaccurate comment regarding these studies. We have revised the sentence to focus on the different purposes of these studies.

17. P4L27, I don't understand how the authors defined 'datasets'. A paired observation or a published experiment? Response: Thank you for your comment. The term datasets revers to sets of paired observations. We have revised the text to state 'paired observations.'

18. P5L19, what are the chemical properties included in this study? Please list all of them here. This is a good place to explain how 'C/N/P remaining' or 'C/N/P release' were calculated. In addition, does 'wood' mean woody tissues or leaves of woody species? Response: Thank you for your comment. Following your suggestion, we have added information on the relevant chemical properties and the calculation of 'C/N/P remaining' and 'C/N/P release'. In the previous version of the manuscript, 'wood' referred to the leaves of woody species; we have replaced this term with 'leaf source (forest or grassland)' in the revised manuscript.

19. Results, direct and indirect effects of UV radiation represent fundamentally different processes. One involves exposing litter to various levels of UV radiation, while the other refers to growing plants under UV manipulation and then accessing decay processes.

Figure 2 is the only place where the direct and indirect were differentiated, but I find it impossible to understand. I have the impression that the direct and indirect effects were mixed together in most figures, except in Figure 2. I strongly believe that mixing the two is wrong. Response: Thank you for your comment. The direct and indirect effects of UV radiation on litter decomposition has been studied by Song et al. (2013a). Our study mainly focused on the effects of UV enhancement or attenuation on litter mass loss and nutrients rather than on the direct and indirect effects of UV radiation. In addition, dividing the effects into direct and indirect effects would have reduced the sample size in the analyses, increasing the potential uncertainty in the meta-analysis. Thus, we pooled the direct and indirect effects of UV radiation in most of the figures.

20. In all figures, it's not clear how sample size was counted. Does it include the number of replicates in original studies? Response: Thank you for your comment. The sample size was calculated as the number of paired observations; it does not include the number of replicates in the original studies. We have added this information to the methods section of the revised manuscript.

21. Figure 1, are the results about the 'direct effects' only? What does the error bar represent? It is more intuitive to show 95% confidence intervals with error bars. Response: Thank you for your comment. Fig. 1 shows the results for both the direct and indirect effects. The values in the figure are the weighted response ratios, and the error bars represent standard error. Black symbols indicate significant differences (p < 0.05) between the response ratios and zero.

22. Figure 1, I find it very surprising that there were less published studies on C remaining than on P remaining. Response: Thank you for your comment. There were fewer published studies addressing C remaining than addressing P remaining among the included studies in this meta-analysis.

23. Figure 2, I am having a hard time understanding this figure. Were soil and plant supposed to represent direct and indirect effects, respectively? Response: Thank you

for your comment. 'soil' and 'plant' represent direct and indirect effects, respectively, in this figure. We have added this information to the figure legend for clarity.

24. P7L16, please consider merging UV-B and UV-(A+B) results in the figure 1. Response: Thank you for your suggestion. We merged the UV-B and UV-(A+B) results in Fig. 1 and have moved the UV-B and UV-(A+B) results to the supporting information.

25. P8L1, you mean 'results from field experiments'? Response: Thank you for your suggestion. We have replaced 'A field experiment' with 'The results from field experiments.'

26. Figure 4, if one experiment lasted exactly 4 months, was it categorized as '2-4' or '4- 6' months? Please add text in the figure to cue readers about UV amendment vs. Attenuation. Response: Thank you for your comment. When one experiment lasted exactly 4 months, we categorized as 2-4 months. We have added information regarding the categorization to the section 'Data analysis' and have added information regarding UV enhancement and attenuation to the figure.

27. Figure 4, there are not enough data over 500 mm precipitation to draw reliable conclusion Response: Thank you for your comment. We have presented the results as bars instead of in a regression plot in Fig. 4, as proposed by Reviewer #1, and we have written the description of the results in the manuscript.

28. P8L25-26, use of stats is questionable. The authors need to compare the slopes with 1 statistically. Response: Thank you for your suggestion. We have added the results of statistical analyses of differences in nutrient release and mass loss between UV enhancement or attenuation and ambient environment. The corresponding text has been revised to 'Various effects of changes in UV radiation on the RRs of remaining nutrients and weight remaining were found (Fig. 6). The slope of the RRs of remaining C and N and the weight remaining under UV attenuation were 1.31 and 1.23, respectively, however, the effects of both UV enhancement and UV attenuation on the relationship between each of C, N and P and mass loss relative to the ambient environment were

not significant (p>0.05). Interestingly, UV enhancement significant promoted the lignin release compared with the ambient environment (p<0.01).'

29. P9L5-9, a summary of key results will be more useful to start the discussion. Response: Thank you for your suggestion. The original sentence was redundant to information presented in the introduction section. Following your suggestion, we have revised the beginning of the discussion section to 'In the present study, a meta-analysis was performed to assess the effects of UV exposure on the dynamics of litter decomposition and nutrient release. We found that leaf sources (grassland or forest), experimental condition (field or laboratory), experimental duration, and exposure type (direct or indirect effects) affected litter decomposition and nutrient release under UV exposure.'

30. The discussion overall does not offer much original interpretation of data. In many places, it repeats the results or cites the findings from literature (e.g., P9L20-25, P9L6-10). Response: Thank you for your comment. Follow your comments and those of Reviewer #1 comments, we have carefully revised the manuscript. We have added information, including some explanations of results, such as 'UV enhancement promoted litter decomposition, mainly due to the enhancement of photodecomposition as well as to the high initial litter N content; litter decay showed a significant relationship with N concentration (Figs. S6 and S7).'

31. P9L14-15, this should be presented in the results. Response: Thank you for your suggestion. We have moved the sentence to the results section.

32. P10L20, you mean no effects were observed in the lab? Response: Thank you for your comment. Our intended meaning is that no effects were observed in the lab. We have added this information to the revised manuscript.

33. P11L7, Figure 4 presents the data in an interesting way. I consider it a stronger component of the manuscript. However, it is debatable whether the temporal patterns clearly show three different stages. For example, the UV attenuation results seem

to follow a two-stage pattern. Response: Thank you for your comment. Mass loss showed three different stages over decomposition time under UV enhancement. During the early stage (0-4 months), UV enhancement did not impact mass loss or nutrient release. However, UV enhancement significantly promoted mass loss during the intermediate stage (4-18 months). This result indicated that UV enhancement can accelerate litter decomposition given a sufficient period of UV accumulation, consistent with the results of Wang et al. (2017). However, the UV attenuation results seemed to follow a two-stage pattern, with a significant reduction in mass loss during the early stage; the effect diminished as the decomposition time increased. We have revised the text to 'In the present study, litter decomposition also varied with decomposition time under UV enhancement and attenuation; the effects of UV enhancement on decomposition exhibited three-stage temporal dynamics (Fig. 4). UV enhancement did not impact mass loss during the early stage (0-4 months) but significantly promoted litter decomposition during the intermediate stage (4-18 months). These results indicate that UV enhancement can accelerate litter decomposition given a sufficient period of UV accumulation (Wang et al., 2017) as well as accelerate nutrient release. However, UV attenuation significantly reduced litter decomposition during the early stage, and the effect strength diminished as the decomposition time increased.'

Please also note the supplement to this comment:
https://www.biogeosciences-discuss.net/bg-2018-445/bg-2018-445-AC2-supplement.zip
* * *